# A Comparative Investigation of Mechanical Properties of TiB$_2$/Cr Multilayer Film by Indentation

Simeng Chen [1], Zhengtao Wu [1,*] and Qimin Wang [1,2,*]

1   School of Electromechanical Engineering, Guangdong University of Technology, Guangzhou 510006, China
2   Key Laboratory of Green Fabrication and Surface Technology of Advanced Metal Materials Anhui University of Technology, Ministry of Education, Maanshan 243002, China
*   Correspondence: ztwu@gdut.edu.cn (Z.W.); qmwang@gdut.edu.cn (Q.W.)

**Abstract:** Alternating TiB$_2$-dcMS and Cr-HiPIMS layers are used to fabricate TiB$_2$/Cr multilayer films. Introducing a 5-nm-thick Cr interlayer deposited under a substrate bias of $-60$ V produces slight increases in both film hardness and elastic modulus. The TEM observation indicates that the Cr grains favor epitaxial growth on the TiB$_2$ interlayer, forming a coherent TiB$_2$/Cr interface. This improves hardness. Mechanic measurement by using AFM illustrates that the coherent interface increases the elastic modulus of the Cr up to ~280 GPa, which is significantly higher than bulk material.

**Keywords:** TiB$_2$/Cr; multilayer; mechanical properties; coherent interface

## 1. Introduction

Multilayer films focused on nitride-based materials attract great interest as superhard films and wear-resistance applications. Film properties including hardness, toughness, adhesive strength, and wear resistance are enhanced by incorporating a nanocrystalline multilayer structure [1–5]. These multilayered structures consist of repeating interlayers of two different materials with nanometer-scale thicknesses. Compared to monolayer films, nanomultilayer films have superior mechanical properties. For example, Sun et al. [6] prepared TiAlN/TiB$_2$ multilayer films and found that all multilayer films with well-defined interfaces showed greater hardness than the individual TiAlN and TiB$_2$ layers. It was reported that the increase in hardness was not only due to the laminar structure and its different shear moduli, but also due to the mismatch of lattice constants at adjacent interfaces, which creates tensile stress fields and compressive stress fields, and the forces generated by the two materials with different shear moduli impede the dislocation motion when the dislocation crosses the coherent interface [7]. In general, theories including the interface coordinated strain theory, the coherent epitaxy theory, the Hall-Patch reinforcement effect, and the interface composite theory have been used to demonstrate the mechanism behind hardness improvement [8–11].

TiB$_2$ has been widely used for wear parts, seals, cutting tools, and metal matrix composites due to its high hardness and wear resistance [12]. Multilayering TiB$_2$ with metallics (such as Cr [13], Ti [14], and FeMn [15]), carbides and nitrides (such as TiAlN [16], TiN [17], TiC [18,19], VC [20], and BN [21]), oxides (such as Al$_2$O$_3$ [22]), and carbon-based layers [23] has been reported to further improve the mechanical properties of the TiB$_2$. In our recent work [24], alternating TiB$_2$-dcMS and Cr-HiPIMS layers were used to fabricate TiB$_2$/Cr multilayer films with varying Cr interlayer thickness, 2 and 5 nm, and the substrate bias during the growth of Cr interlayers from floating, to $-60$ V and $-200$ V. The results reveal that increasing the substrate bias during the Cr interlayer growth from floating to $-60$ V produces increases in both film hardness and elastic modulus. However, the atomic-scale microstructure of the TiB$_2$/Cr interface is still unknown. The impact of the interface on film hardness enhancement and the evolution of the mechanical properties of the Cr interlayer is not clear either.

Therefore, an analytical TEM study was performed to obtain a structural insight into the TiB$_2$/Cr interface in this work. Mechanical property measurements of the TiB$_2$/Cr multilayer film by nanoindentation were applied using a constant load mode and the continuous stiffness method (CSM). In addition, the film cross-section was mechanically characterized using AFM (atomic force microscopy). A comparative investigation of the mechanical properties of the TiB$_2$/Cr multilayer film by indentation was conducted.

## 2. Materials and Methods

A hybrid configuration of HiPIMS (high power impulse magnetron sputtering) (CemeCon AG, Würselen, Germany) and dcMS (direct current magnetron sputtering) (CemeCon AG, Würselen, Germany) was employed to fabricate the TiB$_2$/Cr multilayer film. A schematic diagram of the configuration is illustrated in Figure 1a. The Cr target was operated in HiPIMS mode and dc power was applied to the TiB$_2$ targets (Cr-HiPIMS/TiB$_2$-dcMS). Prior to deposition, the 15 × 15 mm$^2$ Si(001) and WC-Co (6 wt% Co) substrates were successfully cleaned in acetone and isoacetone. During deposition, the total system pressure was 3 mTorr (0.4 Pa) and the substrates were heated using a resistance heater with a temperature of 500 °C, measured by a thermocouple mounted next to the substrate. To ensure better adhesion of the TiB$_2$ coating, a 30 nm Cr buffer layer was first deposited on the substrate by HiPIMS. The Cr-HiPIMS power was constant at 1.5 kW (200 Hz, 50 μs pulse length, duty cycle 1%), with a deposition time of 30 s. However, the TiB$_2$-dcMS- power was stable at 2.0 kW, with a deposition time of 5 min, alternated in turn for twenty cycles. A detailed description of the process parameters is presented in Table 1. A 5-nm-thick Cr interlayer deposited under a substrate bias of −60 V was selected to fabricate the multilayer. A schematic diagram of the multilayer structure is illustrated in Figure 1b. Figure 1c presents the θ-2θ XRD pattern of the TiB$_2$/Cr multilayer film. The film has a hexagonal crystal structure characteristic of TiB$_2$ (JCPDS No. 35-0741).

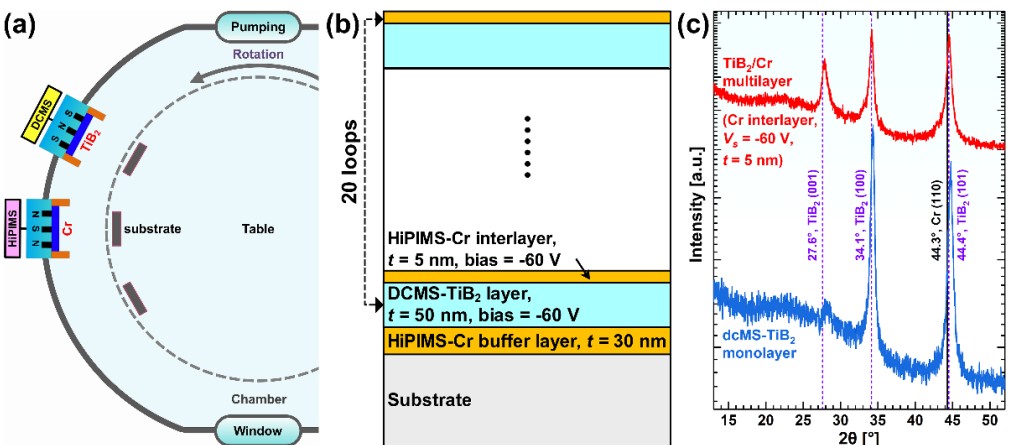

**Figure 1.** Schematic diagrams of (**a**) Cr-HiPIMS/TiB$_2$-dcMS sputtering deposition system and (**b**) TiB$_2$/Cr multilayer. (**c**) presents θ-2θ XRD patterns of TiB$_2$/Cr multilayer film.

**Table 1.** Detail parameters for deposition of TiB$_2$/Cr multilayer film.

| Process Parameters | Values |
|---|---|
| Cr-HiPIMS power [kW] | 1.5 (200 Hz, 50 μs pulse length, duty cycle 1%) |
| TiB$_2$-dcMS power [kW] | 2.0 |
| Deposition pressure [Pa] | 0.4 |
| Deposition temperature [°C] | 500 |
| Bias [V] | −60 (synchronized to the Cr-HiPIMS pulse, offset time 40 μs, pulse length 100 μs) |

The film crystal structure was detected using a Philips Panalytical X'Pert PRO X-ray diffraction system (XRD, D8 Advance Diffractometer, Bruker AXS, Karlsruhe, Germany) operated with Cu $K_\alpha$ ($\lambda$ = 0.154 nm) radiation. The scanning range was 20° to 80° with a scanning step of 0.02° and a dwell time of 0.1 s/step. Film morphologies were investigated using cross-sectional scanning electron microscopy (XSEM Nano 430, FEI Company, Eindhoven, The Netherlands) operated at 10 kV, and cross-sectional transmission electron microscope (XTEM Tecnai $G^2$ F20, FEI Company, Eindhoven, Netherland) with 200 kV accelerated voltage analyses. A Berkovich diamond tip (Ø 20 µm) in an Anton Paar TriTec UNHT$^3$ (Anton Paar, Graz, Austria) instrument was used to determine the mechanical properties of the film. Both the constant load mode under a constant loading of 10 mN and the continuous stiffness method with the max load of 20 mN and amplitude of 10% were used to determine the hardness (H) and elastic modulus (E) of the multilayer film as a function of the maximum load and penetration depth. In addition, the elastic modulus of the film cross-section was characterized using AFM in a Park NX20 machine (Park Systems, Suwon, Korea). It performs on the following principles: a micro-cantilever, which is extremely sensitive to weak forces, is fixed at one end and contains a small needle tip (diamond material) at the other end, which maintains gentle contact with the sample surface. This force causes the cantilever to deflect due to the very weak repulsive forces between the atoms at the tip of the needle and the atoms on the sample surface. The laser beam generated by the laser diode is focused through the lens onto the back of the cantilever and then reflected back to the photodiode to form the feedback. As the sample is scanned, the sample moves slowly on the carrier table and the micro-cantilever is adjusted by the feedback adjustment system to bend and undulate in response to the surface profile of the sample. The reflected beam is then deflected and the surface profile and mechanical information are recorded by the detector.

## 3. Results and Discussion

Figure 2 shows XSEM and XTEM images of the surface and cross-section of the $TiB_2$/Cr multilayer film. The film surface is smooth. Both XSEM and XTEM images indicate that the 1.1-µm-thick multilayer film is dense without cracks or pores in the film cross-section. The $TiB_2$/Cr interface appears sharp and the $TiB_2$ interlayers have a columnar structure. Figure 2e deduced that low-angle (5°) grain boundaries were formed between the $TiB_2$ and the Cr nanograins. The Cr grains favor epitaxial growth on the $TiB_2$ interlayer, forming a coherent $TiB_2$/Cr interface.

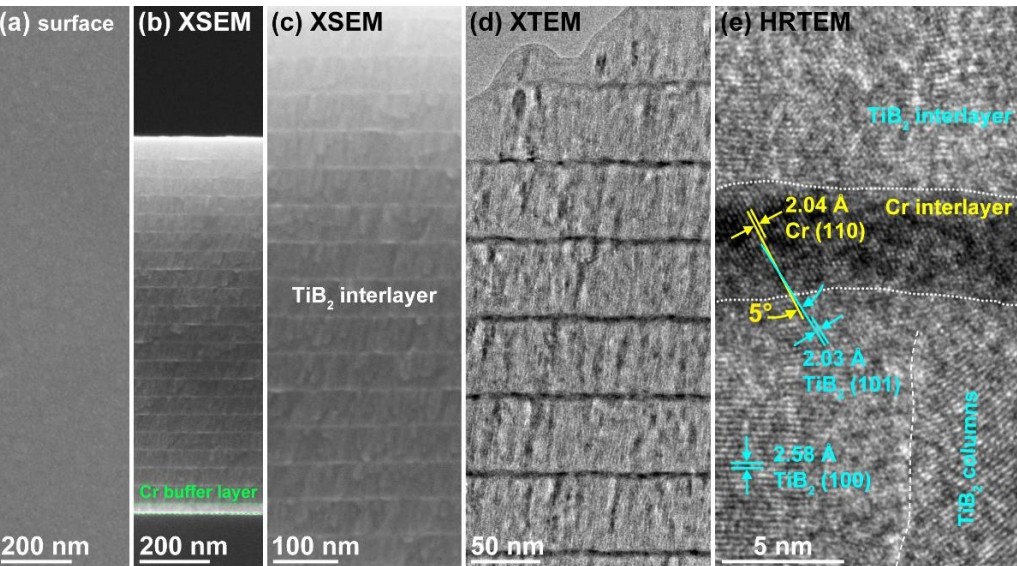

**Figure 2.** (**a**) SEM surface, (**b**,**c**) SEM cross-section, (**d**) XTEM, and (**e**) HRTEM images of the $TiB_2$/Cr multilayer film.

Figure 3a exhibits load-penetration depth curves of nanoindentation tests (constant load mode, max. load 10 mN) applied on the film surface. The maximum penetration depth of the indenter was ~110 nm (~10% of film thickness). The film hardness and elastic modulus were $28.8 \pm 0.8$ and $395 \pm 12$ GPa, respectively. A dcMS-TiB$_2$ monolayer has a hardness of $27.9 \pm 0.8$ GPa and an elastic modulus of $340 \pm 6$ GPa [24]. Introducing a 5-nm-thick Cr interlayer deposited under a substrate bias of $-60$ V produces increases in both film hardness and elastic modulus. Figure 3b presents the load versus time curve of the penetrated indenter using the continuous stiffness measurement mode. The maximum load was 20 mN and the amplitude was ~10% of the real-time load. Figure 3c shows H and E values determined by the CSM mode measurement as a function of the penetration depth. A continuous increase in hardness and decrease in elastic modulus occurred as the penetration depth increased. This could be due to an indentation size effect whereby the measured hardness increases with increasing indentation load [25]. The effect has been attributed to a variety of factors, including the elastic recovery of the indentation, surface dislocation pinning, dislocation nucleation, deformation band spacing, surface energy of the test specimen, and statistical measurement errors. Residual stress may also influence measured hardness values.

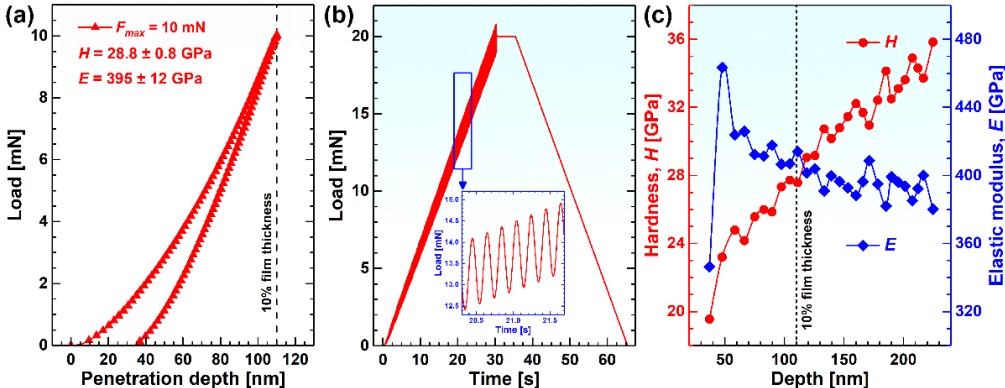

**Figure 3.** (**a**) Load-penetration depth curves of nanoindentation tests applied on the film surface (constant load mode, max. load 10 mN). (**b**) Load vs. time curve of the penetrated indenter using the continuous stiffness measurement mode. (**c**) *H* and *E* values determined by using the CSM mode measurements as a function of the penetration depth.

We attempted to calculate the average grain sizes of these films using TiB$_2$ (100) XRD peaks based on the well-known Scherrer equation. The equation is $D_{hkl} = K \times \lambda/(B_{hkl} \times \cos\theta)$, where $D_{hkl}$ is the crystallite size in the direction perpendicular to the lattice planes, hkl are the Miller indices of the planes being analyzed (TiB$_2$ (100) XRD peaks were used in this study), K is a numerical factor frequently referred to as the crystallite-shape factor, $\lambda$ is the wavelength of the X-rays, $B_{hkl}$ is the width (full-width at half-maximum) of the X-ray diffraction peak in radians, and $\theta$ is the Bragg angle. The results are listed in Table 2. We can see that the Cr layer introduction had a slight impact on the average grain sizes of TiB$_2$. In addition, a negligible change in $2\theta$ values is present in the multilayer film compared to the monolayer film, which indicates a slight change in the stress state (Figure 1c). Therefore, we can conclude that the film stress state has a slight impact on the film hardness.

**Table 2.** Average grain sizes of dcMS-TiB$_2$ monolayer and TiB$_2$/Cr multilayer films.

| Samples | Cr Layer Thickness [nm] | Cr Substrate Bias [V] | $B_{100}$ [°] | Average Grain Sizes of TiB$_2$ [nm] |
|---|---|---|---|---|
| TiB$_2$ monolayer | 0 | - | 0.45 | 15.7 |
| TiB$_2$/Cr multilayer | 5 | $-60$ | 0.46 | 15.1 |

In addition, it was noticed that the Cr grains favor epitaxial growth on the $TiB_2$ interlayer, forming a coherent $TiB_2$/Cr interface. This produces the hardness improvement compared to a monolayer as well. Mechanical characterization using AFM was conducted to investigate the evolution of the mechanical properties of the Cr interlayer, as shown in Figure 4. Figure 4a shows 50-nm thickness of the $TiB_2$ interlayer and 5-nm thickness of the Cr interlayer. Figure 4b shows the AFM morphology of the polished film cross-section. An alternating $TiB_2$/Cr multilayer structure was noticed since the relatively soft Cr layer was much easier to remove compared to the hard $TiB_2$ layer. The measurement by AFM shows that the elastic modulus of the Cr was ~280 GPa, which is significantly higher than the magnetron-sputtered Cr monolayer (~165 GPa [26,27]). Thus, the coherent $TiB_2$/Cr interface increases the mechanical properties of both the $TiB_2$ and the Cr interlayers.

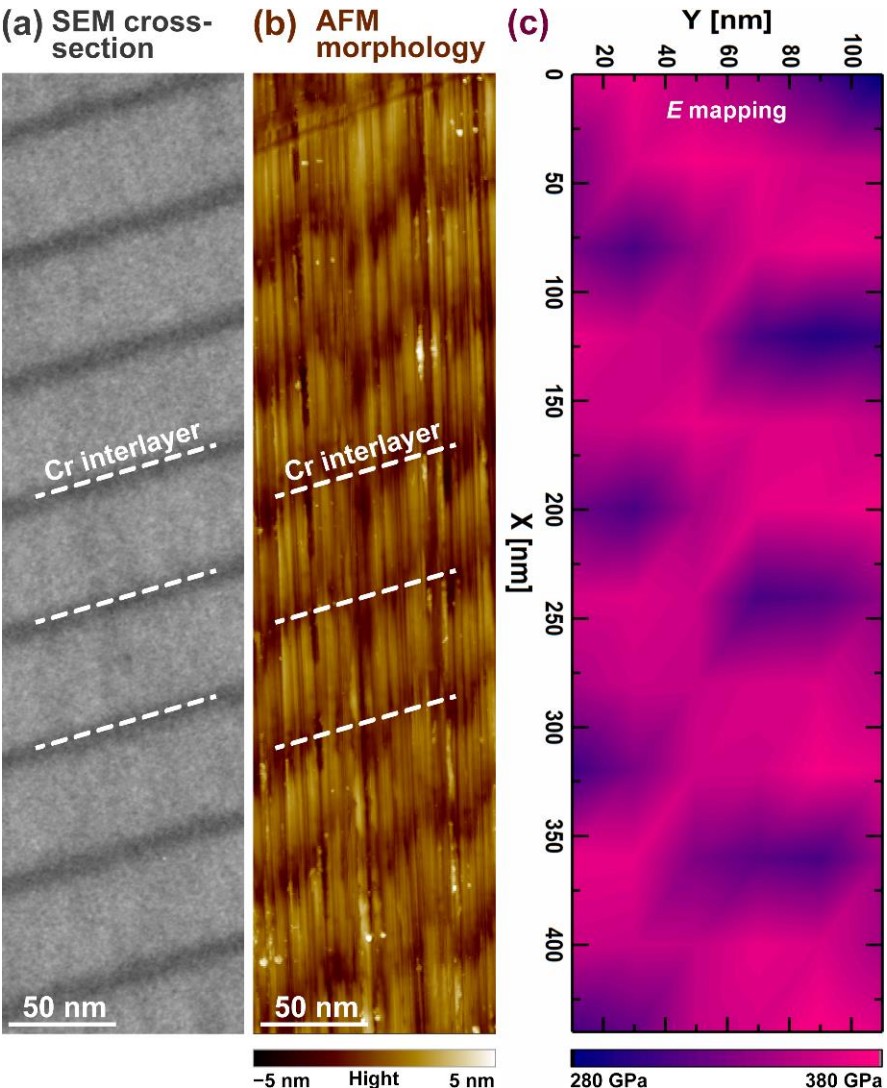

**Figure 4.** Elastic modulus characterization of the $TiB_2$/Cr multilayer film using AFM. (**a**) Cross-section morphology of the film. (**b**) Height of the film cross-section. (**c**) Elastic modulus *E* mapping of the film cross-section.

## 4. Conclusions

$TiB_2$/Cr multilayer film was fabricated using Cr-HiPIMS/$TiB_2$-dcMS hybrid deposition. The Cr layer introduction has a slight impact on the average grain sizes of $TiB_2$. The Cr grains favor epitaxial growth on the $TiB_2$ interlayer, forming a coherent $TiB_2$/Cr interface. This produces a change in the stress state (from tension for monolayer to compressive for

multilayer) and induces a slight increase in film hardness. The hardness and the elastic modulus of the $TiB_2$/Cr multilayer film were $28.8 \pm 0.8$ and $395 \pm 12$ GPa, respectively. Introducing a 5-nm-thick Cr interlayer deposited under a substrate bias of $-60$ V produces increases in both film hardness and elastic modulus compared to those of a dcMS-$TiB_2$ monolayer because of the coherent interface and compressive stress state. The coherent interface results in the strengthening of the Cr interlayer as well, which has a high elastic modulus of ~280 GPa.

**Author Contributions:** Conceptualization, Z.W. and Q.W.; methodology, S.C.; software, Z.W.; validation, S.C., Z.W. and Q.W.; formal analysis, S.C.; investigation, S.C.; resources, Z.W.; data curation, Z.W.; writing—original draft preparation, S.C.; writing—review and editing, Z.W.; visualization, Z.W.; supervision, Z.W.; project administration, Z.W.; funding acquisition, Q.W. All authors have read and agreed to the published version of the manuscript.

**Funding:** This research was funded by the National Key Research and Development Project of China (2017YFE0125400) and the National Natural Science Foundation of China (51901048). This work was partly supported by the open project of the Key Laboratory of Green Fabrication and Surface Technology of Advanced Metal Materials (GFST2020KF06).

**Institutional Review Board Statement:** Not applicable.

**Informed Consent Statement:** Not applicable.

**Data Availability Statement:** Not applicable.

**Acknowledgments:** The authors gratefully acknowledge the financial support of the National Key Research and Development Project of China (2017YFE0125400) and the National Natural Science Foundation of China (51901048). This work was partly supported by the open project of the Key Laboratory of Green Fabrication and Surface Technology of Advanced Metal Materials (GFST2020KF06).

**Conflicts of Interest:** The authors declare no conflict of interest.

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
