# Peer review of "A Comparative Investigation of Mechanical Properties of TiB2/Cr Multilayer Film by Indentation"

_magnetochemistry, doi:10.3390/magnetochemistry8110148_

Round 1

Reviewer 1 Report (Previous Reviewer 3)

Once reviewed the work entitled “A comparative investigation on mechanical properties of TiB2/Cr multilayer film by indentation”, it is possible to appreciate that the authors carried out the suitable corrections to improve their manuscript. The corrections made enrich this research work and allow a clearer appreciation of its scope and its contribution, mainly in superhard films and wear-resistance applications. The corrections carried out are sufficient for this manuscript to be published in this Journal.

Author Response

We thanks the reviewer for this nice comment.

Reviewer 2 Report (Previous Reviewer 2)

The presentation of this study is not well organized with crispy scientific reasoning. Though the study is not innovative, the authors may have some results that are appealing. There are many missing points for claims made by authors. These claims overkill the purpose of the experimental data. The whole manuscript needs to be rewritten thoroughly.    

1.      As stated in the manuscript that  Cr grains favor epitaxially growth on the TiB2 interlayer, forming a coherent TiB2/Cr interface. This therefore can enhance the hardness. I think the enhancement is mainly due to shear deformation or stresses because the thickness is only 5 nm. This could be seen in Fig. 3(b) where the process of penetration exhibits an oscillation during loading. Can authors elaborate more on the relation between oscillatory loading and shear deformation/stress with added citations?

       The comment is to suggest a table for the process parameters. It’s a reference for readers from the research community for future studies. It seems no such revisions were provided.

There is no response for the related shear deformation on the indentation. It is also important that this indentation is coupled with oscillations. The authors did not discuss the impact of this coupling to the accuracy of measurement although a reference was added. Some explanations based on physics for this measurement is important to justify the experimental results.

Another point claimed by the authors is about the induced compression for multilayer structure to increase the hardness. This argument really needs some ways of proof, whether directly or indirectly. For example, if compression in the Cr layer could lead to slightly increased hardness in multilayers, then it must be thickness-dependent. The number of layers and thickness together should be investigated further. In addition, the tension-to-compression transition should be an important issue worthy of some preliminary studies.

Equations for the calculation should be provided with all parameters in order to convince readers of the data shown in Table 1.

2.      Please add a table for the process parameters (power, bias, gas supply, etc.) for the deposition by HiPIMS and DC sputtering. This is important credential information.

     The comment is to suggest a table for the process parameters. It’s a reference for readers from the research community for future studies. It seems no such revisions were provided.

3.      Peak locations (2θ) in XRD should be explicitly marked. Particular attention should be addressed to how to distinguish between Cr (110) and TiB2 (101) in order to be accordant with the analysis from HRTEM.

The authors’ responses do not provide a clear explanation for how to distinguish Cr (110) and TiB2 (101) in XRD. The issue is that there isn’t an observable peak in XRD for Cr(110) in XRD because it’s too thin.

4.      The manuscript may need a native speaker to improve its writing and correct grammatical/spelling errors. 

English writing really needs improving for a journal publication. 

Author Response

  1. As stated in the manuscript that Cr grains favor epitaxially growth on the TiB2 interlayer, forming a coherent TiB2/Cr interface. This therefore can enhance the hardness. I think the enhancement is mainly due to shear deformation or stresses because the thickness is only 5 nm. This could be seen in Fig. 3(b) where the process of penetration exhibits an oscillation during loading. Can authors elaborate more on the relation between oscillatory loading and shear deformation/stress with added citations? The comment is to suggest a table for the process parameters. It’s a reference for readers from the research community for future studies. It seems no such revisions were provided.

There is no response for the related shear deformation on the indentation. It is also important that this indentation is coupled with oscillations. The authors did not discuss the impact of this coupling to the accuracy of measurement although a reference was added. Some explanations based on physics for this measurement is important to justify the experimental results.

Another point claimed by the authors is about the induced compression for multilayer structure to increase the hardness. This argument really needs some ways of proof, whether directly or indirectly. For example, if compression in the Cr layer could lead to slightly increased hardness in multilayers, then it must be thickness-dependent. The number of layers and thickness together should be investigated further. In addition, the tension-to-compression transition should be an important issue worthy of some preliminary studies.

Equations for the calculation should be provided with all parameters in order to convince readers of the data shown in Table 1.

Response: We thanks the reviewer for this comment. We are sorry for resubmitting an old version of revised-manuscript (in which many modifications (reply to Reviewers 2 and 3) are missing). Please check our updated modifications.

  1. i) First, a CSM-mode (continuous stiffness mode) ( Res. Symp. Proc. 522, 1998, pp. 53-64.) was used in the indentation measurement show in Fig. 3(b). An oscillating loading was applied to measure H and E values as a function of the penetration depth.
  2. ii) We added more discussion on hardness improvement in revised manuscript.

TiB2/Cr multilayer film was fabricated using Cr-HiPIMS/TiB2-dcMS hybrid deposition. The Cr layer introduction has slight impact on average grain sizes of TiB2. The Cr grains favor epitaxially growth on TiB2 interlayer, forming a coherent TiB2/Cr interface. This produces the change in the stress state (from tension for monolayer to compressive for multilayer) induces a slight increase of film hardness. The hardness and the elastic modulus of the TiB2/Cr multilayer film were 37.8 ± 1.8 and 407 ± 13 GPa, respectively. Introducing a 5-nm-thick Cr interlayer deposited under a substrate bias of -60 V produces increases of both film hardness and elastic modulus compared to those of a dcMS-TiB2 monolayer because of the coherent interface and compressive-stress state. The coherent interface results in the strengthening of Cr interlayer as well, which has a high elastic modulus of ~280 GPa.

We try to calculate the average grain sizes of these films using TiB2 (100) XRD peaks. The results are listed in Table 1. We can find that the Cr layer introduction has slight impact on average grain sizes of TiB2. In addition, a slight shift to the lower 2θ values is present in multilayer film with respect to the monolayer film which indicates a change in the stress state (Fig. 2b). Based on the fact that the multilayer films’ peak match the bulk value, one may assume that the reference TiB2 film is in slight state of tension. The hardness of the reference TiB2 monolayer film is slightly improved by Cr interlayer introduction. Hardness increases from 27.9 ± 0.8 GPa for TiB2 monolayer to 28.8 ± 0.8 GPa for the TiB2/Cr film with 5 nm Cr interlayer deposited at -60 V bias (Fig. 3a). Therefore, we can make conclusion that the change in the stress state (from tension to compressive) induces a slight increase of film hardness (reply to the comment “I think the enhancement is mainly due to shear deformation or stresses because…”).

  1. Please add a table for the process parameters (power, bias, gas supply, etc.) for the deposition by HiPIMS and DC sputtering. This is important credential information. The comment is to suggest a table for the process parameters. It’s a reference for readers from the research community for future studies. It seems no such revisions were provided.

Response: We thanks the reviewer for this important comment. We add more information in materials and methods. In addition, a table for the process parameters wad added.

  1. Peak locations (2θ) in XRD should be explicitly marked. Particular attention should be addressed to how to distinguish between Cr (110) and TiB2 (101) in order to be accordant with the analysis from HRTEM.

The authors’ responses do not provide a clear explanation for how to distinguish Cr (110) and TiB2 (101) in XRD. The issue is that there isn’t an observable peak in XRD for Cr(110) in XRD because it’s too thin.

Response: We marked peaks locations in revised Fig. 1c. In addition, Figs. 2d and e show bright field TEM images of the TiB2/Cr multilayer film. The heavier Cr atoms have a dark contrast compared to TiB2 (Fig. 2d). Thus, we can distinguish between Cr and TiB2 interlayers in Fig. 2e. The Cr (110) grains favor epitaxially growth on the TiB2 (101) interlayer (Cr (110)//TiB2 (101)) since they have a similar interplanar spacing.

  1. The manuscript may need a native speaker to improve its writing and correct grammatical/spelling errors.

English writing really needs improving for a journal publication.

Response: We thanks the reviewer for this important comment. We do our best to improve the language.

Reviewer 3 Report (New Reviewer)

The research presented in the article is part of the Interface Engineering strategy covering the substrate / coating interface and the interfacial areas between the layers in multilayer coatings, which is essential to enhance and to maintain the performance of protective coatings on metallic substrates. In general, the Interface Engineering strategy includes surface engineering, the selection of the appropriate interlayer between the substrate and the coating, the selection of appropriate, in terms of chemical composition and thickness, layers in the repeating double-layer module of the multilayer coating. In order to interpret hardness improvement in multilayer coatings as compared to single-layer coatings, the interface coordinated strain theory, coherent epitaxy theory, Hall-patch reinforcement effect, and interface composite theory are taken into account, among others.

The research presented  concerned multilayer TiB2 / Cr coatings deposited using various methods, i.e. DC magnetron sputtering was used in the case of TiB2 and HIPIMS in the case of Cr layers, and different thicknesses of the latter were used. The research was focused on the understanding of the TiB2 / Cr intreface and tidal microstructure on the atomic scale and the impact of the interface on film hardness enhancement and the evolution of the mechanical properties of the Cr interlayer.

The research results obtained by the authors on the basis of the application of a rich set of complementary, modern research methods make a significant contribution to the interface engineering strategy, i.e. the influence of the interface area on the change of the mechanical properties of multilayer coatings.

Author Response

We thanks the reviewer for this nice comment!

Round 2

Reviewer 2 Report (Previous Reviewer 2)

This manuscript has been back and forth several times although the quality and content are not appropriately improved due to its original structures.

I re-posted some comments from the 2nd round review. It is strongly recommended that authors address each question with short and clear answers in the manuscript.  

1. There is no response for the related shear deformation on the indentation. It is also vital that this indentation is coupled with oscillations. The authors did not discuss the impact of this coupling on the accuracy of measurement although a reference was added. Some explanations based on physics for this measurement are essential to justify the experimental results.

2. Another point claimed by the authors is about the induced compression for multilayer structures to increase the hardness. This argument needs some ways of proof, whether directly or indirectly. For example, if the reduction in the Cr layer could lead to slightly increased hardness in multilayers, then it must be thickness-dependent. The number of layers and thickness together should be investigated further. In addition, the tension-to-compression transition should be an important issue worthy of some preliminary studies.

3. Equations for the calculation should be provided with all parameters to convince data shown in Table 1.

4. Peak locations (2θ) in XRD should be explicitly marked. Particular attention should be addressed to how to distinguish between Cr (110) and TiB2 (101) to be accordant with the analysis from HRTEM. The authors’ responses do not provide a clear explanation for how to distinguish Cr (110) and TiB2 (101) in XRD. The issue is that there isn’t an observable peak in XRD for Cr(110) in XRD because it’s too thin. 

5. English writing needs to be improved for a journal publication. For example, in the caption of Fig. 3 and 4, it may be better to avoid using itemized terms (a) or (b) as the subjective for a  sentence.

6. Please provide full-width (page-wide) and high-resolution figures or images for fig. 1, 2, and 3. 

Author Response

1. a) There is no response for the related shear deformation on the indentation. b) It is also vital that this indentation is coupled with oscillations. The authors did not discuss the impact of this coupling on the accuracy of measurement although a reference was added. Some explanations based on physics for this measurement are essential to justify the experimental results.

Response: We thank the reviewer for these important comments. Reply to comments:

a) First, we agree that the shear deformation has a significant impact on the indentation tests. In this work, we can’t evaluate the shear deformation since the mechanical properties of Cr layers and TiB2/Cr coherent interface are difficult to be determined. In addition, the thicknesses of the Cr layers are nonuniform. The shear deformation is hard to be evaluated even by using finite element analysis. Thus, we analyzed the indentation results using the method of Oliver & Pharr ( Mater. Res., 1992, 7, 1564-1583) and take no account of the shear deformation.

b) Second, a CSM-mode (continuous stiffness mode) was used in the indentation measurement shown in Fig. 3(b). During the CSM-mode test, an oscillating loading (nonlinear load) was artificially set and applied to measure H and E values as a function of the penetration depth. The oscillations were used to measure H and E values as a function of the penetration depth. They are not mistakes or deviations. A normal mode with applying a linear load can be used to obtain H and E values with a certain maximum-penetration depth.

Therefore, we believe that our indentation measurements and results are reliable. We hope that the above reply will meet your approval.

2. a) Another point claimed by the authors is about the induced compression for multilayer structures to increase the hardness. This argument needs some ways of proof, whether directly or indirectly. b) For example, if the reduction in the Cr layer could lead to slightly increased hardness in multilayers, then it must be thickness-dependent. The number of layers and thickness together should be investigated further. In addition, the tension-to-compression transition should be an important issue worthy of some preliminary studies.

Response: We thank the reviewer for these important comments. Reply to comments:

a) We claim that both average grain sizes and film stress state have slight impacts on the film hardness evolution.

We try to calculate the average grain sizes of these films using TiB2 (100) XRD peaks. The results are listed in Table 2. We can find that the Cr layer introduction has a slight impact on the average grain sizes of TiB2. In addition, a negligible change in 2θ values is present in multilayer film compared to the monolayer film which indicates a slight change in the stress state (Fig. 1c). Therefore, we can conclude that the film stress state has a slight impact on the film hardness.

b) We agree that “if the reduction in the Cr layer could lead to slightly increased hardness in multilayers, then it must be…”. It’s absolutely right.

Actually, in our previous work (Surf. Coat. Technol. 2022, 436, 128337), alternating TiB2-DCMS and Cr-HiPIMS layers are used to fabricate TiB2/Cr multilayer films with varying the Cr interlayer thickness, 2 and 5 nm, and the substrate bias during growth of Cr interlayers from floating, to -60 V and -200 V. The effects of multilayer structure on mechanical properties, static oxidation, and tribological behavior of the TiB2/Cr multilayers are investigated.

In this work, analytical TEM and AFM investigations were performed to obtain a structural insight into the TiB2/Cr interface. The impacts of numbers of layers and thickness are not included.

3. Equations for the calculation should be provided with all parameters to convince data shown in Table 1.

Response: We thank the reviewer for this important comment.

Equations for the calculations of Average grain sizes (Table 2) were added in the revised manuscript (Table 1 indicates detailed parameters for the deposition of TiB2/Cr multilayer film and no equations were used).

To be honesty, the instrumental broadening effect on the XRD calculation result has been calibrated, whereas the influence of the residual stress was still unclear. Both finer grain size and increased microstrain (or internal stress) will result in the broadening of the XRD characteristic peaks. However, determination of the grain sizes by the well-known Scherrer equation does not deal with the microstrain. Therefore, there is a tendency to underestimate the grain sizes because of ignorance of the microstrain effect, particularly for bombarding deposition which produces higher internal strain.

Since a negligible change in 2θ values is present in multilayer film compared to the monolayer film which indicates a slight change in the stress state (Fig. 1c). Therefore, we can conclude that the film stress state changes slightly. Therefore, we can calculate the average grain sizes of films by the well-known Scherrer equation.

4. a) Peak locations (2θ) in XRD should be explicitly marked. b) Particular attention should be addressed to how to distinguish between Cr (110) and TiB2 (101) to be accordant with the analysis from HRTEM. The authors’ responses do not provide a clear explanation for how to distinguish Cr (110) and TiB2 (101) in XRD. The issue is that there isn’t an observable peak in XRD for Cr(110) in XRD because it’s too thin.

Response: We thank the reviewer for these important comments. Reply to comments:

a) We have marked all peak locations in revised Fig. 1c.

b) Yes, there isn’t an observable peak in XRD for Cr(110) in XRD because it’s too thin. In Fig. 1c, we provide theoretical peak locations based on the JCPDS. We didn’t claim that we can observe the characteristic peak of Cr(110). We provide the Cr label just want to let the readers understand the XRD patterns easily.

5. English writing needs to be improved for a journal publication. For example, in the caption of Fig. 3 and 4, it may be better to avoid using itemized terms (a) or (b) as the subjective for a sentence.

Response: We thanks the reviewer for this important comment again. We do our best to improve the language.

6. Please provide full-width (page-wide) and high-resolution figures or images for fig. 1, 2, and 3.

Response: We attached an independent word file of figures.

This manuscript is a resubmission of an earlier submission. The following is a list of the peer review reports and author responses from that submission.

Round 1

Reviewer 1 Report

The manuscript presents and structural characterization of TiB2/Cr multilayered coating deposited high-power-impulse-magnetron sputtering. Authors determined by AFM that TiB2/Cr multilayered coatings possess.

Unfortunately, I consider the manuscript unsuitable for publication in present form by the following reason: All the results included in manuscript are presented in a recently published papers by the same authors in:

Zhengtao Wu, Rongli Ye, Babak Bakhit, Ivan Petrov, Lars Hultman, Grzegorz Greczynski

Reprint of: Improving oxidation and wear resistance of TiB2 films by nano-multilayering with Cr

Surface and Coatings Technology, Volume 442, 25 July 2022, Pages 128602

https://doi.org/10.1016/j.surfcoat.2022.128337

Zhengtao Wu, Rongli Ye, Babak Bakhit, Ivan Petrov, Lars Hultman, Grzegorz Greczynski

Improving oxidation and wear resistance of TiB2 films by nano-multilayering with Cr

Surface and Coatings Technology, Volume 436, 25 April 2022, Pages 128337, https://doi.org/10.1016/j.surfcoat.2022.128602

Reviewer 2 Report

1.      As stated in the manuscript that  Cr grains favor epitaxially growth on the TiB2 interlayer, forming a coherent TiB2/Cr interface. This therefore can enhance the hardness. I think the enhancement is mainly due to shear deformation or stresses because the thickness is only 5 nm. This could be seen in Fig. 3(b) where the process of penetration exhibits an oscillation during loading. Can authors elaborate more on the relation between oscillatory loading and shear deformation/stress with added citations?

2.      Please add a table for the process parameters (power, bias, gas supply, etc.) for the deposition by HiPIMS and DC sputtering. This is important credential information.

3.      Peaks locations (2θ) in XRD should be explicitly marked. Particular attention should be addressed to how to distinguish between Cr (110) and TiB2 (101) in order to be accordant with the analysis from HRTEM.

4.      The manuscript may need a native speaker to improve the writing and correct grammatic/spelling errors.  

Reviewer 3 Report

In sputtering processes it is necessary to include information, despite already being previously reported, such as substrate used, power used, working pressure for each material, temperature and deposition time.

For characterization techniques, it is necessary to give more details in the measurements, for example, type of detector used, working distance, etc.

Further support and evidence is needed to say that multilayer film is dense and smooth.

Explain how the grain size, the phase found and its efforts influence the hardness values, with respect to the TiB2 film, since everything is attributed only to the incorporation of Cr.

Explain how the mechanical characterization was carried out by means of AFM, since what is shown is only the topography of the cross section.

If the thickness of the Cr layer changes and the value of the substrate bias of -60 V is modified, surely the hardness value will continue to change, favoring better values ​​than the TiB2 monolayer layer. In this sense, the conclusions may need to be reinforced, to give greater justification for the increase in mechanical properties, for example, using calculations of the relative volume fraction of the ceramic.